# The Influence of Vaccination of Broiler Chickens and Turkeys with Live *E. coli* Attenuated Vaccine on *E. coli* Population Properties and TRT Vaccination Efficacy

**DOI:** 10.3390/ani11072068

**Published:** 2021-07-11

**Authors:** Marcin Śmiałek, Joanna Kowalczyk, Andrzej Koncicki

**Affiliations:** Department of Poultry Diseases, Faculty of Veterinary Medicine, University of Warmia and Mazury, Oczapowskiego 13, 10-719 Olsztyn, Poland; welencasia@gmail.com (J.K.); koncicki@uwm.edu.pl (A.K.)

**Keywords:** broiler chickens, broiler turkeys, *E*. *coli* vaccination, *E*. *coli* antibiotic susceptibility, TRT vaccination

## Abstract

**Simple Summary:**

*Escherichia coli* infections are considered one of the major causes of economic loss in the poultry industry. The reasons for the magnitude of the problem are the numerous sources of infection with these bacteria for birds and the need for an effective prevention method. Vaccination is one of the strategies for minimizing the consequences of *E. coli* infection. In this study, we performed three independent experiments at farm level using a live vaccine against *E. coli.* Antibiotic-free broiler chickens, conventional broiler chickens and broiler turkeys were examined in different experiments. The most meaningful results and conclusions of these experiments are that vaccination against colibacillosis decreases the population count of *E. coli,* increases the antibiotic susceptibility of field *E. coli* isolates and has no impact on the efficacy of vaccination against another significant poultry upper respiratory tract disease—TRT. We believe that the vaccination of broiler chickens and turkeys against *E. coli* can improve bird health and should be considered in terms of routine immunoprophylaxis.

**Abstract:**

Colibacillosis is one of the major causes of economic losses in the poultry industry. Vaccination against *E. coli* is attracting increasing interest. The aim of the study was to evaluate the influence of vaccination with live, *aroA* gene-deleted vaccine on the structure and properties of field *E. coli* population and its potential impact on TRT vaccination efficacy in broiler chickens and turkeys. We performed three independent experiments on farms: (1) with antibiotic-free broiler chickens, (2) with conventional broiler chickens and (3) with broiler turkeys. In experiment 1, we have recorded an approx. 0–15% prevalence of multi-susceptible *E. coli* strains in the first production cycle. Starting from production cycle number two, after vaccination introduction, successive significant increases in *E. coli* susceptibility emerged, reaching 100% of strains at the end of production cycle 3. Increased *E. coli* susceptibility remained for three production cycles after vaccination withdrawal. In experiments 2 (2 production cycles) and 3 (1 production cycle), we recorded similar tendencies of *E. coli* susceptibility profile change. In experiments 1 and 2, the *E. coli* population count was lower after vaccination. In experiments 2 and 3, no negative influence of *E. coli* vaccination on the level of specific antibodies against TRT was recorded.

## 1. Introduction

Avian pathogenic *Echerichia coli* (APEC) is a causative agent of poultry colibacillosis. The disease has been one of the major causes of economic losses in the poultry industry worldwide. These losses are due to mortality and decreased productivity of affected birds [1]. *Escherichia coli* infections are usually secondary disruptors of homeostasis where that is primarily triggered by other diseases, immunosuppression or disorders in respiratory system functioning [2]. The *E. coli* serotypes which are considered to be particularly pathogenic are O1, O2, O78, O8, and O35 and it is these which are most often isolated from infected birds. The transmission of infection is usually horizontal, with the main source being the contaminated housing environment [2].

The prevention of colibacillosis is problematic, especially given that no effective method has been developed so far. Colibacillosis prevention is largely a matter of good sanitation, hygiene, and elimination of the sources of infection [1], as well as negation of factors that increase the risk of its development in a flock [3].

Recently, an effective commercial vaccine based on the O78:K80 *E. coli* strain with *aroA* gene deletion has been developed and used in the field. Its efficacy has been proven in several studies, particularly in chickens but also in turkeys. The vaccine has been demonstrated to decrease the risk of mortality and development of colibacillosis-related lesions in chickens and turkeys after APEC experimental infection when administered on the first or fifth day of life [4,5]. Although the highest protection was observed against the homologous infection (O:78), to some extent the vaccine also induced cross-protection against other *E. coli* serotypes [4]. Our previous study [6] demonstrated that the vaccination of broiler chickens with this vaccine increased their production profitability. Vaccination against *E. coli* has been shown not to interfere with the effectiveness of vaccinations against other poultry diseases, such as IB (infectious bronchitis), IBD (infectious bursal disease), or ND (Newcastle disease) [6,7].

Another issue associated with poultry APEC infections is the multi-drug resistant (MDR) and extended-spectrum beta-lactamase (ESBL)-producing *E. coli* strains, which are increasingly commonly isolated from poultry colibacillosis cases [8,9,10]. These strains can pose a direct risk to consumers since they can be passed to humans via the food chain or by direct contact, and they can also pass antibiotic resistance genes to other bacteria [11].

Our previous study [6] demonstrated that spray vaccination of day-old broilers against *E. coli* in the sixth week of life decreased the number of isolates of the bacteria from internal organs excluding those of the respiratory system and that the *E. coli* strains isolated from the vaccinated birds were more susceptible to antimicrobials. These results indicate that vaccination can decrease the risk and level of infection and that improve antibiotic susceptibility of field *E. coli* strains.

Considering the need for prevention of colibacillosis and the supposed inimicality of *E. coli* vaccination to MDR, a study was undertaken under field conditions to verify the results of previous studies addressing the influence of *E. coli* vaccination on a decreased field *E. coli* population count and the improvement of their antimicrobial susceptibility. The study was carried out with three experimental models, which were antibiotic-free broiler chickens, conventional broiler chickens, and broiler turkeys. It additionally evaluated the influence of *E. coli* vaccination on the efficacy of simultaneous vaccination against avian metapneumoviruses (aMPV) in chickens and turkeys.

## 2. Materials and Methods

### 2.1. Ethic Statement

According to information from the Local Ethics Committee in Olsztyn, no special approval was necessary for the experiments performed under field conditions. All animal procedures, vaccination, and sample collection were performed during standard veterinary inspections and observations.

### 2.2. Experimental Layout

#### 2.2.1. Experiment 1

Experiment 1 was carried out on farm 1, operating only one chicken house (CH1) with the capacity of 35,000 broiler chickens. It was an antibiotic-free farm, on which no therapeutic antibiotics had been used for at least two years prior to the experiment. Hence, experiment 1 entailed no use of antibiotics. It was staged over six subsequent production cycles. In the first three (production cycles 1–3), all birds on were vaccinated against colibacillosis. The vaccination was performed on the farm on the first day of life immediately after they had arrived at CH1 in transport boxes. The chicks were vaccinated against *E. coli* with a coarse spray at a dose recommended by the producer and observed for 15 min. Afterwards, they were allowed out of the boxes onto litter. Vaccination against *E. coli* was not continued in the last three production cycles (production cycles 4–6). The chicks were also vaccinated at the hatchery against IB using a mixture of the Mass-like and 793B strains with a coarse spray throughout the entire experiment. Over the six production cycles, swabs were taken for microbiological analyses from the internal organs of 3 randomly selected birds in the third and sixth weeks of life. At each sampling, swabs from three birds were preserved for analyses. The samples were collected immediately after diagnostic euthanasia of the birds under sterile conditions using dry swabs without transport media. Peritoneum, liver, lungs, heart, jejunum, hip and hock joint, trachea, and suborbital sinus swabs were taken. Individual swabs were transferred for microbiological analyses with semiquantitative determination of the *E. coli* colony count, and the isolated *E. coli* strains were evaluated for their susceptibility to a panel of 20 antimicrobials.

#### 2.2.2. Experiment 2

Experiment 2 was carried out on farm 2 on which two chicken houses were in use (CH2 and CH3) with capacity of approximately 25,000 birds each. The birds from CH_2_ were vaccinated against *E. coli* on the 10th day of life with a coarse spray, at a dose recommended by the producer. Birds from CH_3_ served as the control. The birds from both chicken houses were vaccinated against IB at the hatchery with the Mass-like and 793B strain mixture (common vaccination schedule against IB), and also against TRT using Clone K of aMPV subtype A, at a dose recommended by the producer. The house 2 birds were vaccinated against *E. coli* and TRT with one preparation made by mixing both vaccines in one diluent. Experiment 2 was continued for two subsequent production cycles. In weeks 3 and 6 of life in the first production cycle and in week 6 of life in the second production cycle, swabs were collected for microbiological analyses from the internal organs of three randomly selected birds each from CH2 and CH3. The swab types and sampling method were the same as those described in experiment 1. During both production cycles, 23 blood samples were additionally taken from birds kept in CH2 and CH3 in week 6 of life for serological analyses aimed at determining the level of specific antibodies against aMPV.

#### 2.2.3. Experiment 3

Experiment 3 was carried out on a two-turkey-house farm of male broiler turkeys with approximately 3500 birds in each house. This experiment was performed during one production cycle. The turkeys from turkey house 1 (TH1) were vaccinated against *E. coli* twice, i.e., on the first day and in the third week of life. The vaccination procedure was the same as in experiments 1 and 2. All of the birds were also vaccinated against TRT using Clone K of aMPV subtype A at a dose recommended by the producer. In TH1, the vaccination against *E. coli* and TRT on the first day and in the third week of life was administered in a mixture of both vaccines in one diluent. In week 3 and 6 of life, samples were collected for microbiological analyses from the birds’ internal organs as in experiments 1 and 2. In addition, in weeks 10 and 14 of life, swabs were collected from the nasal cavity, trachea, and cloaca of six live birds from both turkey houses for microbiological analyses. In weeks 6 and 14 of life, 23 blood samples were additionally taken from birds kept in both turkey houses for serological analyses aimed at determining the level of specific antibodies against aMPV.

### 2.3. Antimicrobial Use on Experimental Farms

In each experiment, antimicrobials were not administrated to birds for at least 10 days after *E. coli* vaccination. In experiment 1, no antimicrobials were used throughout the entire study. In experiments 2 and 3, the antimicrobial treatment scheme was the same for both houses in individual experiments (CH2 and CH3 in experiment 2 and TH1 and TH2 in experiment 3).

### 2.4. Birds and Vaccination

In experiments 1 and 2, chicken houses were populated with Ross 308 broiler chicks of both sexes purchased from one hatchery (which changed between experiment 1 and experiment 2) one hatch, and one reproduction flock.

In experiment 3, turkey houses were settled with male B.U.T. 6 turkey poults, purchased from one hatchery and from one hatch. If the source of the eggs was one reproduction flock or several was unascertainable.

A live, attenuated, *aroA* gene-deleted *E. coli* vaccine (Zoetis, Madison, NJ, USA) was used in all experiments. The vaccine comprises the viable *E. coli* O78 strain. The vaccine strain from the same batch as the vaccines used in all experiments was analyzed for antimicrobial susceptibility with the same method used on the isolated field strains. Feed and water were given to the birds ad libitum in both experiments.

### 2.5. Microbiological Studies

Microbiological studies were performed as described previously [6]. Briefly, after 24 h of pre-incubation in tryptic soy broth (Argenta, Poznań, Poland) at 40.5°C, further incubation proceeded at the same temperature for 24 h on Columbia agar with 5% addition of defibrinated sheep blood and MacConkey agar media (Argenta, Poznań, Poland). After morphological and biochemical evaluation, *E. coli* isolates were evaluated for their susceptibility to a panel of 20 antimicrobials, according to the Clinical and Laboratory Standards Institute guidelines [12]. The following antimicrobials were analyzed: amoxycillin (amx), amoxycillin + clavulanic acid (amc), ceftiofur (cef), clindamycin (cli), colistin sulfate (cst), doxycycline (doxy), enrofloxacin (enr), erythromycin (ery), gentamycin (gen), florfenicol (ff), flumequine (fluq), lincomycin/spectinomycin (ls), marbofloxacin (mbf), neomycin (nm), norfloxacin (nor), oxytetracycline (otc), penicillin G (pen), sulfamethoxazole/trimethoprim (sxt), tetracycline (tet), and tylosin (tyl) (Argenta, Poznań, Poland). Susceptibility was assessed on a scale of 0–3, where 0 denoted no susceptibility. The strains that were sensitive to 35% (7 or more) of the antimicrobials tested with a susceptibility level of at least 1 were referred to as multisensitive strains, as described previously [6].

The results of the microbiological analyses are presented as the mean number of *E. coli* isolates from internal organs or swabs of vaccinated or control birds at different periods of the experiment. The results of antimicrobial susceptibility testing are presented as the mean percentage of multisensitive strains among all *E. coli* isolated strains at different periods of the experiment. Additionally, panel testing results are presented as the mean number of antimicrobials the isolated *E. coli* were susceptible to.

### 2.6. Serological Evaluation

A commercial ELISA APV Ab Test kit (IDEXX Laboratories, Westbrook, MN, USA) was used to determine the titer of TRT-specific IgY in broiler and turkey serum in experiments 2 and 3, respectively. Individual stages of the tests were performed with an epMotion 5075 LH liquid handler (Eppendorf, Hamburg, Germany), an EL × 405 automatic multi-well plate washer (BioTek Instruments, Winooski, VT, USA), and an EL × 800 plate reader (BioTek Instruments). The mean geometric titer of antibodies and CV% were calculated for each group in each sampling period.

### 2.7. Statistical Analysis

Statistical analysis was performed with GraphPad Prism 6.05 (GraphPad Software, San Diego, CA, USA) using the Mann–Whitney *U* test. Differences were considered statistically significant at *p* < 0.05.

## 3. Results

### 3.1. Experiment 1

During the first production cycle in the experiment 1, the mean number of *E. coli* isolates reached 8.67 and 9.00 in the 3rd and 6th weeks of life, respectively. The number of *E. coli* isolates decreased during production cycle 2 but the differences between isolate totals in the first and second production cycles for respective sampling periods were not statistically significant. Starting from production cycle 3, the mean number of *E. coli* isolates from the internal organs of birds decreased significantly and remained significantly lower during production cycles 4–6 in comparison to the first cycle at both sampling periods (week 3 and 6). The results of *E. coli* isolation from experiment 1 bird samples are summarized in Table 1.

The results of antimicrobial susceptibility testing of the isolated *E. coli* strains are summarized in Table 2. During the first production cycle, the mean number of antimicrobials to which the *E. coli* isolates displayed susceptibility reached 5.00 and 2.27 in weeks 3 and 6 of life, respectively. In production cycles 2–3, in each sampling period the mean number of antimicrobials the *E. coli* strains were susceptible to increased significantly (except for production cycle 3 birds in the third week of life) and peaked at the end of this stage reaching 11.0 antimicrobials. In production cycles 4–6, the mean susceptibility decreased slightly in comparison to production cycle 3, but still remained significantly higher than the mean in the first production cycle. At the same time, the percentage of multisusceptible isolates was elevated from production cycle 2 and remained so throughout the entire experiment (Table 2).

### 3.2. Experiment 2

Throughout experiment 2, the mean number of *E. coli* isolates was lower in the vaccinated birds, while the percentage of multisusceptible strains was elevated compared to the percentage of such in the control broilers. In each sampling period of both production cycles, the mean number of antimicrobials the *E. coli* isolates were susceptible to was significantly higher in the vaccinated than in the control birds. The results of microbiological studies from experiment 2 are summarized in Table 3.

The results of serological evaluation of the mean antibody titers against aMPV from experiment 2 are summarized in Table 4. In both production cycles, no differences in the mean anti-aMPV antibody titers were recorded between *E. coli*-vaccinated chickens and control birds.

### 3.3. Experiment 3

The results of microbiological analyses from experiment 3 are summarized in Table 5. In most of the sampling periods of this experiment, the mean *E. coli* isolates count in the vaccinated birds was higher than or equal to the count in the control animals. Starting from the sixth week of the turkeys’ lives, the percentage of multisusceptible strains increased in the vaccinated birds and peaked in the 14th week of life, reaching 72.22% of all isolates. In weeks 6 and 14, a significant increase was recorded in the mean number of antimicrobials the *E. coli* isolates from the *E. coli*-vaccinated turkeys were susceptible to in comparison to this parameter in control turkeys.

The results of serological evaluation of the mean antibody titers against aMPV from experiment 3 are summarized in Table 4. In every sampling period of experiment 3, no differences in the mean anti-aMPV antibody titers were recorded between the *E. coli*-vaccinated and the control birds.

## 4. Discussion

Colibacillosis has been one of the major causes of economic losses in the poultry production. The key points in controlling avian colibacillosis are management interventions, infection controls, and vaccination strategies [2,13].

The number of cases of infections with multi-drug resistant and ESBL-producing *E. coli* has recently increased [8,9,10]. This situation poses a severe threat to consumers because these bacteria can act as donors of antibiotic resistance genes to other bacteria [11].

In the current study no clinical cases of colibacillosis were recorded in none of the experiments and in none of the production cycles. Despite this issue high number of *E. coli* strains were isolated from experimental birds which allow further evaluation of the properties of those strains. Selection and randomization of birds for laboratory analysis were completed based on the body weight of the birds which had to be equal to the mean body weight of the bird in the chicken house, as well as on the clinical condition of the birds selected for laboratory analyses, which had to be representative of the flock (birds displaying a clinical condition that could be interpreted as a reflection of the general clinical condition of birds in a chicken house were selected).

It has been demonstrated that vaccination against colibacillosis contributed to a reduction in the amount of *E. coli* in the population of vaccinated birds [6,14]. To some extent, our study confirms those observations as we noted a similar tendency in the number of *E. coli* isolates in broiler chickens from experiments 1 and 2. In experiment 3, the numbers of *E. coli* isolates were similar in both the vaccinated and the control turkeys. It should be borne in mind, however, that in this experiment, the nasal cavity, trachea, and cloaca swabs were sampled from live birds from week 6 of life. Śmiałek et al. [6] noted fewer isolated *E. coli* bacteria for most of the internal organs, but not for the respiratory system. It was demonstrated previously that after *E. coli* vaccination an immune system in the upper respiratory tract is stimulated which in turn results in the local production of specific IgA and stimulation of CD4^+^ T cells, which was suggested to be the mechanism behind vaccine efficacy in the minimization of *E. coli* spreading to different internal organs from the respiratory tract, after the infection [15]. We cannot exclude that this was also an issue in our study, which would explain those dependencies, however none of the *E. coli* strains from the vaccinated birds possessed the same antimicrobial susceptibility profile as the vaccine strain does. The above-mentioned data suggest that after vaccination of birds under farm conditions the stimulation of upper respiratory tract immune system is efficient enough to minimize the relocation of the field *E. coli* from the gate of the infection (respiratory tract) to other internal organs. This would also explain the results which were obtained in experiment 3 with broiler turkeys.

In our previous study, we have demonstrated that *E. coli* vaccination of broiler chickens contributed to a successive increase in antimicrobial susceptibility of isolated *E. coli* strains [6]. Those findings are consistent with the results of experiments conducted in the present study, including also these obtained for broiler turkeys. The observed phenomenon could be the reason of vaccine and field *E. coli* strains mutual gene transmission which results in the increased antimicrobial susceptibility of the latter. Considering the restoration of the susceptibility to antimicrobials, interesting data were presented for poultry coccidiosis. The introduction of a population of pathogen with a high antimicrobial susceptibility spectrum into resistant population can, contribute to the increased susceptibility of the pathogen field population [16]. This can be achieved by vaccinating the birds with live vaccines containing *Eimeria* spp. [17,18].

Interesting conclusions could be drawn from the results of experiment 1, where those phenomena were investigated in antibiotic-free broiler chickens. As it transpired, even though antibiotics had not been used at this farm for a long time prior to the experiment, highly antibiotic resistant *E. coli* strains were indeed isolated from chickens in the first production cycle. As the vaccination with live *E. coli* vaccine continued, we noted a successive increase in antimicrobial susceptibility of these strains. High susceptibility among *E. coli* isolates was also determined in samples collected during the three consecutive production cycles after vaccination withdrawal, which undoubtedly points to the modulating effect of the vaccination on the field population of *E. coli*. In this study we have differentiated the field *E. coli* isolates from the vaccine one, based on the results of antimicrobial susceptibility profiles, as described previously [6]. Similar to the outcome of our previous study [6], the antibiotic susceptibility profiles of *E. coli* strains isolated from birds from all experiments were different from the susceptibility profile of the vaccine *E. coli* strain (data not shown; the vaccine strain was susceptible to 15 out of 20 antimicrobials tested). Additionally, the vaccine producer states that the *E. coli* strain used in the vaccine can be detected in the cloaca swabs of vaccinated birds for no longer than five weeks after the vaccination and that it is not possible to carry over the strain from one production cycle to a subsequent one. The data given above and the results of our studies exclude the possibility of vaccine bacteria re-isolation in microbiological studies in these experiments. Śmiałek et al. [6] also noted a significantly lower amount of antibiotics used in the flocks of birds vaccinated against colibacillosis. In the present study, the schemes of use and doses of antibiotics (used in experiments 2 and 3) were the same for the vaccinated and the control birds. This was expected to exclude the impact of antibiotic therapy on the ultimate results obtained for *E. coli* population count and properties.

From previous reports, it can be concluded that vaccination against *E. coli* did not affect the immune response to different concurrent viral vaccines such as those against IBD, ND and IB [6,7]. The present study additionally confirmed a lack of such dependencies in the case of vaccination against TRT in both broiler chickens and broiler turkeys. The vaccines based on the subtype A of avian metapneumoviruses are known for their low immunogenicity when stimulating the production of specific post-vaccination antibodies detected with the ELISA test [19,20], which explains why the results of the present study show low titers of post-vaccination antibodies in the examined birds. In our experiment, we found no significant differences in the levels of these antibodies between the control and *E. coli*-vaccinated groups of both chickens and turkeys. Interestingly, considering the often-overlapping scheme of vaccination against TRT and *E. coli* in field veterinary practice, it is worth emphasizing that the vaccines used in the present study were prepared in one diluent and administered to birds simultaneously via a coarse spray. This experimental scheme was expected to mirror the field practices as much as possible.

## 5. Conclusions

From the presented results of the three independent experiments, it can be concluded that vaccination of poultry against colibacillosis with the use of a live, gene-deleted vaccine improves the antimicrobial susceptibility of field *E. coli* isolates, and also it lowers the *E. coli* population count. Vaccination against the bacteria had no negative impact on the efficacy of simultaneous TRT vaccination. Very similar results were recorded with respect to both broiler chickens and turkeys. The mechanisms of modulating the impact of the vaccine *E. coli* strain on the antimicrobial susceptibility of field bacteria population remain unelucidated and definitely merit further studies.

## Figures and Tables

**Table 1 animals-11-02068-t001:** Summarized results of *E. coli* isolation from experiment 1 bird samples.

Production Cycle Number	Bird Age—Week of Life	Mean Number of *E. coli* Isolates ± SD	*p* Value
1	3	8.67 ± 0.58	-
	6	9.0 ± 0.0	-
2	3	7 ± 1.73	0.299
	6	3.0 ± 1.0	0.091
3	3	2.0 ± 0.0 *	0.025
	6	3 ± 2.65	0.059
4	3	3.67 ± 1.53 *	0.037
	6	3.0 ± 1.73 *	0.027
5	3	3.33 ± 1.53 *	0.015
	6	2.33 ± 0.58 *	0.025
6	3	2.0 ± 0.0 *	0.015
	6	2.33 ± 0.58 *	0.003

* Values significantly different in comparison to results from production cycle 1 in the same sampling period. Results were considered statistically significantly different if *p* < 0.5.

**Table 2 animals-11-02068-t002:** Summarized results of *E. coli* antimicrobial susceptibility testing from experiment 1.

Production Cycle Number	Bird Age—Week of Life	Percentage of Multisusceptible Strains ^1^	Mean Number of Antimicrobials to Which *E. coli* Strains Displayed Susceptibility ± SD	*p* Value ^2^
1	3	15.38%	5.0 ± 1.62	-
	6	0%	2.27 ± 0.83	-
2	3	42.86%	6.67 ± 1.15	0.024 *
	6	66.67%	7.11 ± 1.17	<0.0001 *
3	3	33.33%	6.33 ± 0.52	0.111
	6	100%	11.0 ± 1.12	<0.0001 *
4	3	90.91%	8.55 ± 1.81	0.0002 *
	6	66.67%	9.33 ± 2.65	0.0002 *
5	3	90%	8.3 ± 1.64	0.0013 *
	6	42.86%	6.29 ± 2.69	0.0075 *
6	3	100%	9.17 ± 1.47	0.002 *
	6	42.86%	7.57 ± 1.72	<0.0001 *

^1^ if a given *E. coli* strain was susceptible to 7 or more antimicrobials out of 20 tested, the strain was classified as multisusceptible. ^2^
*p* value was calculated based on the mean number of antimicrobials to which the *E. coli* strains displayed susceptibility. * statistically significant difference between the mean number of antimicrobials to which *E. coli* displayed susceptibility, between results for birds of particular production cycles (2–6) and ages (3 or 6 weeks) and birds from production cycle number 1 and of particular ages (3 or 6 weeks). Results were considered statistically significantly different if *p* < 0.05.

**Table 3 animals-11-02068-t003:** Summarized results of microbiological studies from experiment 2.

Production Cycle Number	Bird Age—Week	Chicken House	Mean *E. coli* Isolates Count ± SD	Percentage of Multi-Susceptible Strains ^1^	Mean Number of Antimicrobials to Which *E. coli* Strains Displayed Susceptibility to ± SD	*p* Value ^2^
1	3	2–vaccinated	4.33 ± 2.31	76.92%	8.69 ± 3.01	<0.0001 *
		3–control	6.33 ± 1.15	15.79%	4.42 ± 1.68	-
	6	2–vaccinated	3.33 ± 2.08	90.0%	9.30 ± 2.21	<0.0001 *
		3–control	5.33 ± 1.53	0%	3.06 ± 0.57	-
2	6	2–vaccinated	2.00 ± 0.00	50%	6.17 ± 0.98	<0.0001 *
		3–control	2.67 ± 1.53	0%	2.63 ± 0.52	-

^1^ if a given *E. coli* strain was susceptible to 7 or more antimicrobials out of 20 tested, the strain was classified as multisusceptible. ^2^
*p* value was calculated based on the mean number of antimicrobials to which the *E. coli* strains displayed susceptibility. * statistically significant difference between the mean number of antimicrobials to which *E. coli* displayed susceptibility and between vaccinated and control birds in particular stages of the experiment. Results were considered statistically significantly different if *p* < 0.05.

**Table 4 animals-11-02068-t004:** Results of aMPV serological evaluation in *E. coli* vaccinated and control unvaccinated birds in experiments 2 and 3.

Experiment	Production Cycle	Week	Parameter	Chicken/Turkey House	
				Vaccinated	Not Vaccinated
2	1	6	Gmean	242	312
			CV%	82.8	86.2
	2	6	Gmean	221	241
			CV%	83.6	87.9
3	1	6	Gmean	276	203
			CV%	157.4	131.1
		14	Gmean	391	312
			CV%	78.2	122.2

**Table 5 animals-11-02068-t005:** Summarized results of microbiological studies from experiment 3.

Production Cycle Number	Bird Age—Week of Life	Turkey House	Mean *E. coli* Isolates Count ± SD	Percentage of Multisusceptible Strains ^3^	Mean Number of Antimicrobials to Which *E. coli* Strains Displayed Susceptibility to ± SD	*p* Value ^4^
1	3 ^1^	1—vaccinated	6.33 ± 0.58	0%	3.21 ± 0.98	0.887
		2—control	5.00 ± 2.65	13.33%	3.13 ± 2.10	-
	6	1—vaccinated	3.33 ± 0.58	30%	5.50 ± 1.27	0.027 *
		2—control	3.00 ± 1.00	0%	3.22 ± 1.56	-
	10 ^2^	1—vaccinated	2.84 ± 0.41	11.76%	2.18 ± 2.10	0.084
		2—control	3.00 ± 0.00	0%	1.28 ± 0.46	-
	14	1—vaccinated	3.00 ± 0.00	72.22%	7.61 ± 2.30	<0.0001 *
		2—control	3.00 ± 0.00	0%	2.06 ± 1.47	-

^1^ in weeks 3 and 6 swab samples were collected from 9 internal organs of euthanised birds (*n* = 3). ^2^ in weeks 10 and 14 swabs were collected from nasal turbinates, trachea and cloaca from live birds (*n* = 6). ^3^ if a given *E. coli* strain was susceptible to 7 or more antimicrobials out of 20 tested, the strain was classified as multisusceptible. ^4^
*p* value was calculated based on the mean number of antimicrobials to which the *E. coli* strains displayed susceptibility. * statistically significant difference between the mean number of antimicrobials to which *E. coli* displayed susceptibility and between vaccinated and control birds in particular stages of the experiment. Results were considered statistically significantly different if *p* < 0.05.

## Data Availability

Data is contained within the article. The datasets used and/or analyzed during the current study are available from the corresponding author on reasonable request.

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
