# Peer review of "The Influence of Vaccination of Broiler Chickens and Turkeys with Live E. coli Attenuated Vaccine on E. coli Population Properties and TRT Vaccination Efficacy"

_animals, 2021, doi:10.3390/ani11072068_

Round 1

Reviewer 1 Report

  • Title/Simple summary/Abstract: The english text should be revised. There are errors is grammar, structure and spelling
  • General comment: The English language and structure is poor. The text needs editing and revision and should be checked thoroughly.
  • Materials and methods: The study tries to investigate the protective efficacy of a live E.coli vaccination by its effect in E.coli counts isolated from broiler birds during the production cycle. However, the authors have chosen to evaluate E.coli numbers isolated from birds organs without challenging the birds with E.coli. There is a possibility that the flock is not exposed to the risk of APEC strains. The researchers could have chosen a different protocol by inoculating birds with a fixed concentration of an E.Coli suspension and test for E.coli  re-isolation after vaccination. I would suggest the authors to check relative papers in literature (e.x. An Assessment of the Level of Protection Against Colibacillosis Conferred by Several Autogenous and/or Commercial Vaccination Programs in Conventional Pullets upon Experimental Challenge; Koutsianos et al., 2020).
  • No significant statistical conclusions can be made by using only 3 birds per flock. There is no information about the birds sampled. Did they show any macroscopic lesions of colibacillosis or were chosen randomly? There is no data about the characteristics of the isolated E.coli (serotypes, virulence factors)? Do you know if the strains isolated are APEC or commensal E.coli?
  • Sample size (3 birds per flock) is limited concerning antimicrobial sensitivity testing.
  • The authors have used the term multi-sensitive strains to describe E.coli strains that were sensitive to more than 7 antimicrobial. Can you give some references related to the term multi-sensitive strains?
  • The authors should explain more sufficiently the different results in E.coli re-isolation that were observed in experiment 3.
  • More references could be used in the discussion session in order to support the conclusions

Author Response

Answers to Reviewer 1:

Title/Simple summary/Abstract: The english text should be revised. There are errors is grammar, structure and spelling

Answer: The paper has been corrected by the native speaker in order to meet reviewers recommendations. The confirmation of the correction was implemented in the cover letter.

General comment: The English language and structure is poor. The text needs editing and revision and should be checked thoroughly.

Answer: The paper has been corrected by the native speaker in order to meet reviewers recommendations. The confirmation of the correction was implemented in the cover letter.

Materials and methods: The study tries to investigate the protective efficacy of a live E.coli vaccination by its effect in E.coli counts isolated from broiler birds during the production cycle. However, the authors have chosen to evaluate E.coli numbers isolated from birds organs without challenging the birds with E.coli. There is a possibility that the flock is not exposed to the risk of APEC strains. The researchers could have chosen a different protocol by inoculating birds with a fixed concentration of an E.Coli suspension and test for E.coli  re-isolation after vaccination. I would suggest the authors to check relative papers in literature (e.x. An Assessment of the Level of Protection Against Colibacillosis Conferred by Several Autogenous and/or Commercial Vaccination Programs in Conventional Pullets upon Experimental Challenge; Koutsianos et al., 2020).

Answer: We would like to thank the reviewer for this comment. Unfortunately experimental infection could not be performed in our study, since the entire study took place at a field farm level. Additionally, this was not the aim of our study to evaluate the efficacy of the vaccination against APEC infection. The aim of our study was clearly established in the introduction section and this was further evaluated and discussed in the light of existing papers.

"Considering the need for prevention of colibacillosis and the supposed inimicality of E. coli vaccination to MDR, a study was undertaken under field conditions to verify the results of previous studies addressing the influence of E. coli vaccination on a decreased field E. coli population count and the improvement of their antimicrobial susceptibility."

In order to eliminate the risk of artificial results we have performed studies for a long time period which covered 6 consecutive production cycles in the Experiment number one and two in experiment 2 (as an examples). The obtained results in the experiment, planned as described in our paper, makes it highly unlikely that the observed differences were the reason of a coincidence.

No significant statistical conclusions can be made by using only 3 birds per flock. There is no information about the birds sampled. Did they show any macroscopic lesions of colibacillosis or were chosen randomly? There is no data about the characteristics of the isolated E.coli (serotypes, virulence factors)? Do you know if the strains isolated are APEC or commensal E.coli?

Sample size (3 birds per flock) is limited concerning antimicrobial sensitivity testing.

Answer: Appropriate changes were made in the manuscript body. Differentiation of field and vaccine E. coli strains were performed as described below (part of the modified manuscript):

"In this study we have differentiated the field E. coli isolates from the vaccine one, based on the results of antimicrobial susceptibility profiles, as decribed previously [6]. Similarly to the outcome of our previous study [6], the antibiotic susceptibility profiles of E. coli strains isolated from birds from all experiments were different from the susceptibility profile of the vaccine E. coli strain (data not shown; the vaccine strain was susceptible to 15 out of 20 antimicrobials tested). Additionally, the vaccine producer states that the E. coli strain used in the vaccine can be detected in the cloaca swabs of vaccinated birds for no longer than five weeks after the vaccination and that it is not possible to carry over the strain from one production cycle to a subsequent one."

The number of birds (n=3) collected from one house was determined by two facts: the first one was the budget which allowed us to perform studies the way it was described in experimental layouts. The second one was the suggestion of a statistician who has recommended that in our case the n=3 is enough in the case of Mann Whitney U test (see below).

The birds were chosen randomly. Randomization of samples was ensure by two criteria: (1) the body weight of the birds selected for laboratory analyzes had to be equal to the mean body weight of the bird in the chicken house, (2) clinical condition of the birds selected for laboratory analyzes had to be representative for the flock - only the birds displaying clinical condition that could be interpreted as the one reflecting the general clinical condition of birds in a chicken house, were selected. (appropriate changes has been made in the manuscript body)

"In the current study no clinical cases of colibacillosis were recorded in none of the experiment and in none of the production cycles. Despite this issue high number of E. coli strains were isolated from experimental birds which allow further evaluation of the properties of those strains. Selection and randomization of birds for laboratory analyzis were done based on the body weight of the birds which had to be equal to the mean body weight of the bird in the chicken house, as well as on the clinical condition of the birds selected for laboratory analyzes which had to be representative for the flock (birds displaying clinical condition that could be interpreted as the one reflecting the general clinical condition of birds in a chicken house, were selected). "

This issues also highlight the statistical analysis that has been implemented in our study. The t-test assumes that the compared variables are normally distributed in the two groups. When this assumption is in doubt (for instance, due to the low number of samples analyzed), the non-parametric Mann Whitney U-test is suggested as an alternative.

The authors have used the term multi-sensitive strains to describe E.coli strains that were sensitive to more than 7 antimicrobial. Can you give some references related to the term multi-sensitive strains?

Answe: Corrections were made in the manuscript body in order to meet reviewers recommendation.

"The strains that were sensitive to 35% (7 or more) of the antimicrobials tested with a susceptibility level of at least 1 were referred to as multisensitive strains, as described previously [6]."

The authors should explain more sufficiently the different results in E.coli re-isolation that were observed in experiment 3.

More references could be used in the discussion session in order to support the conclusions

Answer: Discussion was modified in order to meet reviewers recommendations and all modifications made were highlighted in the manuscript body. 4 additional References were implemented in the discussion.

Reviewer 2 Report

I would be glad to review the manuscript if revised with the help of a native English speaker. I have made some edits to the simple summary and the abstract for the author's reference. 

  • Authors have mentioned “E. coli” without space throughout the manuscript.
  • Authors used specific terminology without any introduction, such as Mass-like and 793B (lines:99, 113) strains mixture, aMPV subtype A (lines: 113, 121), so authors need to provide an introduction of what it is. 
  • Lines 29-30: Sentence must be rephrased.

 Simple Summary: Escherichia (E.) coli infections of poultry are considered one of the major causes behind economic losses in the poultry production. This is a reason of for a numerous sources of infection with this bacteria to birds and that, so far, no effective method for its prevention was developed. Vaccination is one of the strategies of for minimizing the consequences of E. coli infection. In this stude study, we've performed three independent experiments on a farm level with the use of using a live vaccine against E. coli. Antibiotic-free broiler chickens, conventional broiler chickens, and broiler turkeys were examined in different experiments. The most meaningful results and conclusions of these experiments is are that vaccination against colibacillosis decreases the population count of E. coli,  it increases the antibiotic susceptibility of field E. coli isolates, and that it doesn't have an and doesn't impact on the vaccination efficacy against other poultry upper respiratory tract disease - TRT. We believe that the vaccination of broiler chikens chickens and turkeys against E. coli can increase birds’ health status should be considered in terms of routine immunoprophylaxis.

  Abstract: Colibacillosis is one of the major causes behind of economic losses in the poultry production. Vaccination against E. coli is gaining increasing interest. The aim of the study was to evaluate the influence of vaccination with live, AroA gene deleted vaccine on the structure and properties of field E. coli population and its potential impact on the efficacy of vaccination against TRT in broiler chickens and turkeys. We've performed three independent experiments on the farms of (1) antibiotic-free broiler chicken, (2) conventional broiler chickens, and (3) broiler turkeys. In experiment 1, we've recorded approx. 0-15% of multi-susceptible E. coli strain in the first production  cycle. From production cycle number two, after vaccination introduction, successive significant increases of E. coli susceptibility emerged, reaching 100% strains at the end of production cycle 3. Increased E. coli susceptibility remained for three production cycles after vaccination withdrawal. In experiments 2 (2 production cycles), and 3 (1 production cycle) we've recorded similar tendencies of E. coli susceptibility profile changes. In experiments 1 and 2, the E. coli population count was decreased after vaccination. In experiments 2 and 3, birds were vaccinated simultaneously against E. coli, and TRT and no negative influence of E. coli vaccination on the level of specific antibodies against TRT was  were recorded.

Author Response

Answers to Reviewer 2

I would be glad to review the manuscript if revised with the help of a native English speaker. I have made some edits to the simple summary and the abstract for the author's reference. 

Authors have mentioned “E. coli” without space throughout the manuscript.

Answer: The paper has been corrected by the native speaker in order to meet reviewers recommendations. The confirmation of the correction was implemented in the cover letter.

Authors used specific terminology without any introduction, such as Mass-like and 793B (lines:99, 113) strains mixture, aMPV subtype A (lines: 113, 121), so authors need to provide an introduction of what it is. 

Answer: Those are the commonly known names of strains and types of infection bronchitis viruses and avian metapneumoviruses, which are commonly used in the vaccination schemes. Changes were made in the manuscript body in order to meet reviewers recommendations.

Lines 29-30: Sentence must be rephrased.

Answer: The paper has been corrected by the native speaker in order to meet reviewers recommendations. The confirmation of the correction was implemented in the cover letter.

Simple Summary: Escherichia (E.) coli infections of poultry are considered one of the major causes behind economic losses in the poultry production. This is a reason of for a numerous sources of infection with this bacteria to birds and that, so far, no effective method for its prevention was developed. Vaccination is one of the strategies of for minimizing the consequences of E. coli infection. In this stude study, we've performed three independent experiments on a farm level with the use of using a live vaccine against E. coli. Antibiotic-free broiler chickens, conventional broiler chickens, and broiler turkeys were examined in different experiments. The most meaningful results and conclusions of these experiments is are that vaccination against colibacillosis decreases the population count of E. coli,  it increases the antibiotic susceptibility of field E. coli isolates, and that it doesn't have an and doesn't impact on the vaccination efficacy against other poultry upper respiratory tract disease - TRT. We believe that the vaccination of broiler chikens chickens and turkeys against E. coli can increase birds’ health status should be considered in terms of routine immunoprophylaxis.

Answer: The paper has been corrected by the native speaker in order to meet reviewers recommendations. The confirmation of the correction was implemented in the cover letter.

Abstract: Colibacillosis is one of the major causes behind of economic losses in the poultry production. Vaccination against E. coli is gaining increasing interest. The aim of the study was to evaluate the influence of vaccination with live, AroA gene deleted vaccine on the structure and properties of field E. coli population and its potential impact on the efficacy of vaccination against TRT in broiler chickens and turkeys. We've performed three independent experiments on the farms of (1) antibiotic-free broiler chicken, (2) conventional broiler chickens, and (3) broiler turkeys. In experiment 1, we've recorded approx. 0-15% of multi-susceptible E. coli strain in the first production  cycle. From production cycle number two, after vaccination introduction, successive significant increases of E. coli susceptibility emerged, reaching 100% strains at the end of production cycle 3. Increased E. coli susceptibility remained for three production cycles after vaccination withdrawal. In experiments 2 (2 production cycles), and 3 (1 production cycle) we've recorded similar tendencies of E. coli susceptibility profile changes. In experiments 1 and 2, the E. coli population count was decreased after vaccination. In experiments 2 and 3, birds were vaccinated simultaneously against E. coli, and TRT and no negative influence of E. coli vaccination on the level of specific antibodies against TRT was  were recorded.

Answer: The paper has been corrected by the native speaker in order to meet reviewers recommendations. The confirmation of the correction was implemented in the cover letter.

Round 2

Reviewer 1 Report

The authors provided sufficient answers to my comments.